# A Novel Cell-Free DNA Fragmentomic Assay and Its Application for Monitoring Disease Progression in Real Time for Stage IV Cancer Patients

**DOI:** 10.3390/cancers17213583

**Published:** 2025-11-06

**Authors:** Sudhir K. Sinha, Hiromi Brown, Kevin Knopf, Patrick Hall, William D. Shannon, William Haack

**Affiliations:** 1Cadex Genomics, Redwood City, CA 94062, USA; patrickhall9636@att.net (P.H.); bhaack@cadexgenomics.com (W.H.); 2InnoGenomics, New Orleans, LA 70148, USA; hbrown@innogenomics.com; 3Sutter Health, Berkeley, CA 94704, USA; kevinbknopf@gmail.com; 4BioRankings, Saint Louis, MO 63108, USA; bill@biorankings.com; 5Division of General Medicine & Geriatrics, Washington University School of Medicine, Saint Louis, MO 63110, USA

**Keywords:** tumor response, cfDNA, therapy monitoring, fragmentomics, retrotransposons, metastatic cancer, liquid biopsy

## Abstract

For patients with advanced (stage IV) cancer, imaging can take 6 to 8 weeks to show whether a new treatment is helping—a stressful delay that can keep them on ineffective therapies. A new blood test has been developed to detect treatment failure in just 2–3 weeks. The test measures tiny tumor-derived DNA fragments in the blood and reports a “Progression Score” from 0 to 100. High scores flag likely rapid cancer growth, while low scores suggest the therapy is effective. Because the test is quick, is noninvasive, and does not rely on specific genetic mutations, it can be used across many cancer types and treatments, helping doctors decide sooner whether to continue, modify, or discontinue a patient’s therapy.

## 1. Introduction

Cell-free DNA (cfDNA) has been widely studied as a cancer biomarker and has been proposed for oncologic applications, including early detection, recurrence monitoring, prognosis, and therapy monitoring [1,2,3,4]. Unfortunately, until recently, unlike circulating-tumor DNA (ctDNA) [5,6,7,8], cfDNA has lacked the specificity required for clinical use. With the advent of fragmentomics as a cancer biomarker, there has been renewed interest in cfDNA in combining cfDNA and fragmentomic signature as a cancer biomarker [9,10,11,12,13,14]. A pioneering machine learning model, DELFI (DNA Evaluation of Fragments for early Inception), has demonstrated high sensitivity and specificity in detecting cancer [15]. Cell-free DNA (cfDNA) from cancer cells exhibits distinct fragmentation patterns compared to healthy cfDNA and has been applied to the early detection of various types of cancers [16,17,18,19]. CfDNA fragments from tumor cells tend to be shorter [20,21,22] and also show characteristic end-motif preference [23]. These unique fragmentomic features have been mainly profiled using whole-genome sequencing [24,25,26]. A more cost-effective, low-coverage whole-genome sequencing approach to capture the fragmentomic pattern has also been utilized [27,28]. The ability to accurately predict the tissue of origin is another utility of the fragmentomic signature [29]. Beyond early detection, whole-genome sequencing-based fragmentomics has also been successfully demonstrated for tracking tumor burden and therapeutic efficacy as a tumor-agnostic method [30,31].

In this paper, a cfDNA fragmentomic assay is proposed, hereafter referred to as the Progression Score (PS) assay, with the analytical sensitivity and clinical specificity to enable clinicians to identify stage IV cancer patients experiencing disease progression. The PS assay is based on quantitative PCR (qPCR) measurements of cell-free DNA (cfDNA) fragments from the high-copy-number retrotransposable element present in patient plasma. Previous studies have reported that the quantification of multicopy retrotransposon fragments in cfDNA and the calculation of the DNA Integrity Index can be correlated with disease progression [32,33]. In the current study, we specifically quantify a very short cfDNA fragment—greater than 80 bp but shorter than 105 bp—using quantitative PCR (qPCR), targeting an ALU element. We then correlate changes in the concentration of this small DNA fragment with disease progression, as assessed by CT scans of solid tumors in patients with colorectal, lung, and breast cancers undergoing treatment.

Disease progression is a commonly used endpoint for determining the efficacy of therapeutic agents. Response evaluation criteria in solid tumors (RECIST 1.1) are a widely used criterion for assessing the status of a patient’s disease for clinical studies [34,35]. In clinical practice, while most physicians do not strictly follow RECIST criteria for determining whether to adjust a patient’s treatment plan, almost all periodically image their patients to assess whether the patient’s disease remains under control. It is generally recognized that imaging for assessing therapy efficacy has weaknesses [36]. For example, some patients with so-called stable disease may have progressive disease, and others may have progressive disease long before imaging can detect tumor growth [37,38]. With the rapid expansion of immunotherapy, clinicians are increasingly recognizing that standard imaging often fails to distinguish between actual progression and treatment-related changes in the early phases of care. Notably, pseudoprogression may mimic tumor enlargement on radiographic studies even when the underlying disease is improving [39]. This is particularly problematic for immunotherapy clinical studies that use disease progression as an endpoint. “Mixed responses” on imaging—where cancer seems to be growing in one area but perhaps stable in another—is another challenging clinical area, as is residual positivity on PET scan. In clinical practice, physicians often face difficult treatment decisions when an image reveals a patient’s disease progression. Clinical decisions can be made more difficult if the patient experiences hyperprogression from immunotherapy [40].

Cancer biomarkers such as CEA, PSA, CA19-9, and CA-125 are often used to monitor patients, but their lack of sufficient clinical specificity makes them inadequate for use in clinical decision-making [41,42]. Up to one-third of patients do not express these tumor markers.

A blood-based assay that quantifies tumor-derived circulating cell-free DNA and demonstrates high clinical specificity for the early detection of disease progression would satisfy a critical unmet need in oncologic care. Earlier insight would enable timely changes to the treatment plan, lessen toxicity from ineffective drugs, lower the cost of futile care, and make better use of therapeutic, hospital, and administrative resources.

## 2. Materials and Methods

Following previously successful proof-of-concept studies [33,43,44] demonstrating the potential of a cfDNA fragmentomic assay to identify, in real time, stage IV cancer patients whose disease has progressed, we prospectively enrolled participants in an observational study to develop a practical assay that could be used by physicians in clinical practice to monitor tumor response in patients under their care.

### 2.1. Study Design

An observational study was designed to prospectively collect blood samples from cancer patients at two time points during treatment. Patients can be enrolled at any point during their treatment, regardless of the treatment plan. Two blood draws were collected. The first blood draw was taken within two days before the infusion of the first treatment cycle, which was administered following the baseline scan. The second blood draw was performed 12 to 21 days after the first treatment and before the next infusion of therapy.

Patients were only enrolled if a CT scan was planned for between 9 and 12 weeks following the first blood draw. Participants were enrolled regardless of which drugs were delivered, irrespective of the line of therapy, and regardless of where they were, even when starting a new line of therapy. The enrollment period for each patient was from the baseline scan to the assessment scan, which was required to be performed between 9 and 12 weeks apart. Board-certified radiologists measured lesions from each scan. Oral medications were taken according to the regular schedule. The flow diagram of the CADEX-0001 study is provided in Figure 1.

### 2.2. Participants

We enrolled 146 stage IV breast, colorectal, and lung cancer patients at 11 sites. For most sites, the study was approved by the Institutional Review Board (IRB) for human subjects at WCG Clinical. For all remaining sites, the study was approved by the respective IRBs of the institution. All study participants provided signed consent for the collection and cfDNA analysis of their blood.

Patients were excluded from the study if they had a secondary malignancy, were being actively treated for autoimmune disease, or had DVT, PE, or sepsis within the past 12 days. Participants were withdrawn from the study if these conditions developed within 12 days of the second blood draw. Of the 146 patients enrolled in the study, 128 were included in the analysis. For various reasons, 18 patients were excluded from the analysis (see Appendix A, which summarizes patients enrolled in the cancer cohort).

### 2.3. Sample Collection and Transportation

One to three peripheral blood samples of 8–10 mL were collected in each Streck^TM^ tube. The specimens were transported at ambient temperature via overnight courier service to Cadex Genomics’ lab in New Orleans. Since the delay in the processing of blood samples affects the concentrations of cell-free DNA [45]. Any participant whose first or second blood draw sample was received at the laboratory more than 120 h after the blood sample was collected was excluded from the study.

### 2.4. Plasma Separation

A two-step centrifugation protocol was used to separate plasma. First, the Cell-Free DNA BCT (Streck) tubes were centrifuged at 1600× *g* for 10 min at 15 °C. Then, the plasma was centrifuged again in a 1.5 mL tube at 16,000× *g* for 10 min at room temperature. The plasma aliquots were transferred to 2 mL cryogenic tubes and stored at −80 °C

### 2.5. Cell-Free DNA Extraction

cfDNA was extracted from 500 µL of plasma using the QIAamp Circulating Nucleic Acid Kit (Qiagen, Germantown, MD, USA) following the kit protocol for a plasma volume of 1 mL with the following modifications: (1) omission of carrier RNA from the ACL buffer, (2) addition of 500 µL of 1 X PBS buffer (Molecular Biologicals International, Inc./Growcells.com, Irvine, CA, USA) to the 500 µL of plasma to increase the sample volume to 1 mL, and (3) extension of the proteinase K digestion time from 30 min to a 1 h incubation. Subsequently, cfDNA was eluted in 60 µL of the kit elution buffer. Each plasma sample was extracted in duplicate. Baseline and after-treatment samples from the same patient were extracted together to avoid batch effects.

### 2.6. Analytical Methods

Two human-specific retrotransposons, Alu Yb8 and SVA [46,47,48,49], were selected as amplification targets to quantify different fragment sizes of cell-free DNA in patient plasma. qPCR primers and probes for each target were designed using the PrimerQuest™ Tool from Integrated DNA Technologies (Coralville, IA, USA). Short (Alu Yb8 = 80 bp, Alu Yb8 = 105 bp) and long (SVA = 265 bp) primers were multiplexed to create two primer mixes: an 80–265 primer mix and a 105–265 primer mix. The SVA primers and probes present in the multiplex serve as both a quality control measure and an enhancement for Alu marker amplification by blocking non-specific amplification. The primer design is schematically represented in Figure 2, which illustrates the multiplex qPCR setup.

Two multiplexes were used for the assay. Multiplex 1 consisted of primers and probes for an 80 bp ALU target, a 265 bp SVA target, and a 172 bp synthetic oligo target. Multiplex 2 consisted of primers and probes for a 105 bp ALU target, a 265 bp SVA target, and a 172 bp synthetic oligo target.

To detect inhibitors in the sample, a 172 bp synthetic nucleotide sequence was used as an internal positive control (IPC) and added to each primer mix. The shift in Ct values of IPC between samples in the same run indicated the presence of inhibitors and served as a quality control parameter. The 265 bp SVA amplicon quantification value was used as a common metric across the two multiplexes for the same sample to flag potential experimental error. The hybridization probe for the short target was labeled with FAM, the long target with Cy5, and the IPC with HEX. All HPLC-purified primers and probes were purchased from Integrated DNA Technologies (Coralville, IA, USA). The primer mixes contain primers and probes for each target (short, long, and IPC) along with PCR enhancer additives. Standard curve assays were conducted on the ABI 7500 or the QuantStudio™ 5 Real-Time qPCR system (Applied Biosystems, ThermoFisher Scientific, Waltham, MA, USA). An approach similar to the multiplexing strategy for the two multicopy retrotransposon targets, ALU and SVA, published earlier [46], was employed for optimizing multiplexes 1 and 2.

Standard DNA was extracted from a single donor’s blood using organic extraction (Proteinase K/SDS digestion, phenol/chloroform extraction, ethanol precipitation, and dissolution in Tris-EDTA buffer—10 mM Tris, 0.1 mM EDTA, pH 8.0) and calibrated against NIST Human DNA Quantitation Standards SRM 2372 Components B (National Institute of Standards and Technology, Gaithersburg, MD, USA). Two microliters (2 μL) of standard DNA or unknown extracted cfDNA were amplified in triplicate in a 20 μL reaction volume, which included 7.7 μL of primer mix, 0.3 μL of the ROX reference standard (diluted to 6 μM), and 10 μL of Brilliant Multiplex QPCR Master Mix (Agilent Technologies). The PCR conditions consisted of one enzyme activation cycle for 10 min at 95 °C, followed by 40 cycles of a 2-step qPCR (15 s at 96 °C and 2 min at 64 °C combined annealing/extension time). DNA samples were quantified using both the 80-265-IPC primer mix and the 105-265-IPC primer mix. qPCR data analysis was performed using the automatic baseline feature of the QuantStudio-5 Design and Analysis Software v1.5.1 (Applied Biosystems, ThermoFisher Scientific, Waltham, MA, USA).

The results of the analytical evaluation of the qPCR multiplex 80-265-IPC are presented in the Appendix A.

### 2.7. Statistical Analyses

To identify predictors of disease progression, we measured four primary variables in each blood sample: SM1 (concentration of >80 bp cfDNA, first blood draw), MM1 (concentration of >105 bp cfDNA, first blood draw), SM2 (concentration of >80 bp cfDNA, second blood draw), and MM2 (concentration of >105 bp cfDNA, second blood draw). From these, four derived variables were calculated: Frag1 = SM1 − MM1 (floor = 0.0), Frag2 = SM2 − MM2 (floor = 0.0), FragDiff = Frag2 − Frag1, and MMDiff = MM2 − MM1.

These eight variables were evaluated in logistic regression models to predict disease progression, defined as an increase in the sum of tumor lesion diameters by ≥20% on CT scans at 8–12 weeks. Based on prior studies demonstrating the association between cfDNA fragment size distributions and tumor progression [32,33], we tested six pre-defined models (Table 1) combining subsets of these variables to capture biologically relevant patterns in cfDNA fragmentation. This hypothesis-driven approach was chosen to leverage established biological insights and minimize the risk of spurious associations associated with data-driven feature selection in a relatively small cohort (*n* = 128). The models were fitted using R version 4.4, and the area under the receiver operating characteristic curve (AUC) was calculated for each. The model with the highest AUC was selected as the preferred model.

To assess robustness and mitigate overfitting, we performed leave-one-out cross-validation (LOOCV), in which each patient was sequentially excluded from model fitting and their outcome was predicted to obtain an unbiased estimate of performance. LOOCV was chosen to maximize the use of available data. The selected model was then further analyzed using the bootstrap method [50] to determine the cut-point for making a progression call.

## 3. Results

### 3.1. Model Selection

Using data from the cancer patient cohort (*n* = 128, see Appendix A), we fitted six pre-defined logistic regression models (Table 1) to predict disease progression, confirmed by CT scans at 8–12 weeks, defined as an increase in the sum of tumor lesion diameters by ≥20%. Radiology reports were analyzed by board-certified radiologists to determine progression status. The area under the receiver operating characteristic (ROC) curve (AUC) was calculated for each model (Table 2).

The model combining FragDiff and MMDiff achieved the highest AUC of 0.934 (*p*-value < 0.001), see Figure 3. The Bonferroni multiple testing adjustment was applied to control the error rate. Leave-one-out cross-validation (LOOCV) yielded an AUC of 0.880, indicating strong discrimination with a modest reduction from the training AUC, consistent with expected optimism correction. This suggests the model is robust, though validation in an independent cohort is planned to confirm generalizability (Section 4.2).

**Figure 3 cancers-17-03583-f003:**
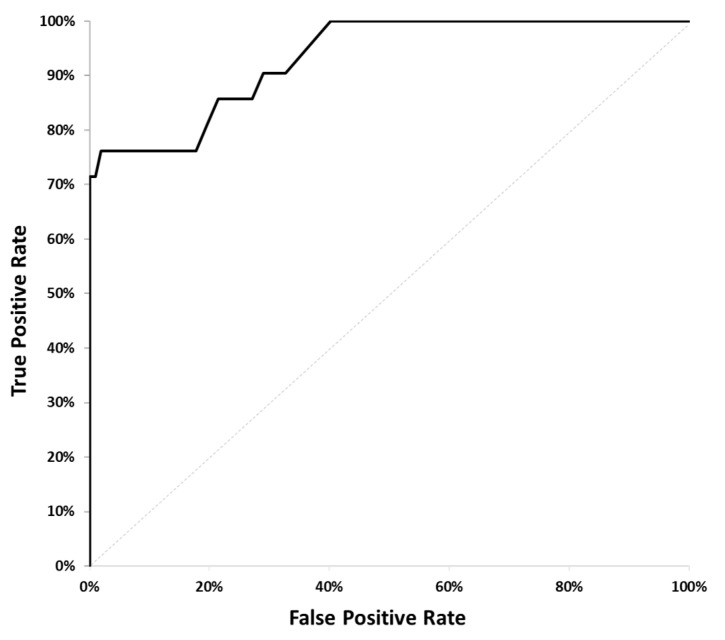
ROC Curve for the FragDiff + MMDiff model.

### 3.2. Progression Score Cut-Point Selection

Following the example of the 21-gene recurrence score assay [51] a progression score was generated with a range of 0 to 100. The model with the highest ROC value was analyzed to determine the appropriate cut-off for making a disease progression call. A cut-off was selected to optimize the assay for specificity and positive predictive value (PPV). Using 1000 iterations of the bootstrap method, the expected PPV for each PS was calculated, and a cut-point with an expected PPV ≥99% was selected. The cut-off selected was 90, see Figure 4.

**Figure 4 cancers-17-03583-f004:**
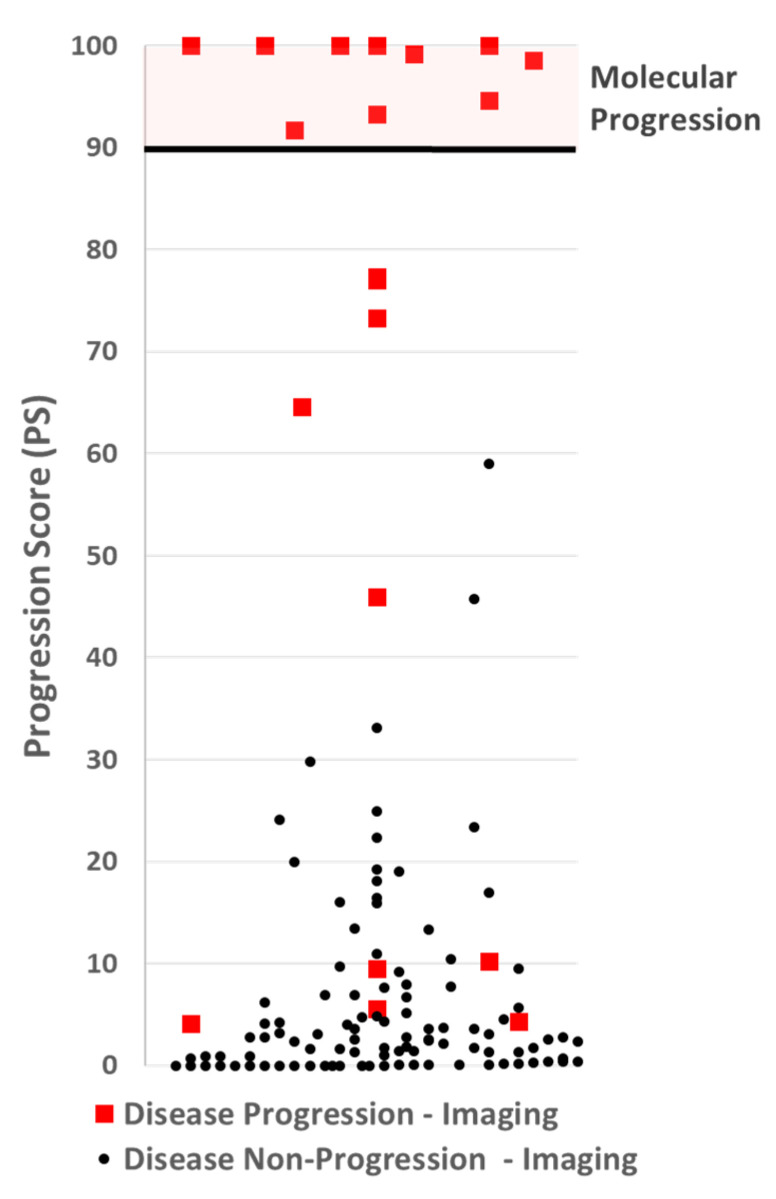
A scatter plot of Progression Scores (PSs). No participants with non-progressive disease by imaging (black dots), had a PS over 60, well below the PS cut-off of 90 for making a molecular progression call. Five participants with progressive disease by imaging had a PS at or below 10.

### 3.3. Assay Performance

By design, a cut-off threshold was set to avoid false positives, making the PPV 100%. Nine (9) patients with disease progression by imaging had a PS below 90, making the negative predictive value 92%. The results are summarized in Table 3.

## 4. Discussion

### 4.1. Role and Limitations of Fragmentomics

Our previous work has shown that cfDNA can reliably identify disease progression and therapy futility [33]. However, that work was conducted under sample handling conditions impractical for a commercial assay, requiring rapid plasma separation.

In that study, plasma was separated at a CLIA lab in an academic medical center within hours of specimen collection. For most oncology clinics, this is not practical and, in many cases, impossible. Consequently, to be clinically viable, a cfDNA-based assay targeting disease progression must be able to accommodate the shipment of blood over several days without the risk of producing false-positive results. Furthermore, the clinical utility of non-response to therapy requires that common conditions unrelated to a patient’s cancer do not result in producing invalid or false-positive results. Fragmentomics plays a crucial role in the PS assay’s ability to minimize the generation of false-positive results. The following data is intended to illustrate this.

Table 4 presents the results for four study participants whose blood samples were received at our laboratory with signs of white blood cell lysis. The first three participants (ID #2022, ID #4019, and ID #8003), when measuring the concentration levels of all cfDNA of the 80 bp amplicon, showed strong evidence of disease progression. However, imaging indicated that none of these patients actually had disease progression. In contrast, participant ID#2021 had a slight decrease in the cfDNA concentration levels of the 80 bp amplicon, suggesting no progression. Yet, imaging revealed that patient ID #2021 was experiencing disease progression. Using the fragmentomic components of the assay as described above, the PS Assay accurately identified the status of all four patients.

While fragmentomics clearly improves the performance of the PS assay, there is evidence that limitations exist in making cfDNA quantitation cancer specific. Several non-cancer clinical conditions are known to increase cfDNA in patient plasma [52,53]. To evaluate the impact of acute clinical conditions on the Progression Score (PS) assay, we measured cfDNA concentrations in plasma from 30 consenting adults who experienced a qualifying health event within 12 days prior to collection. The cohort included patients with acute stroke (*n* = 3), asthma requiring hospitalization (*n* = 2), COPD exacerbation (*n* = 3), diabetic ketoacidosis (*n* = 3), severe inflammatory bowel disease (*n* = 4), myocardial infarction (*n* = 2), severe rheumatoid arthritis (*n* = 5), severe seizure (*n* = 3), and viral infection (*n* = 5). Post-event follow-up sampling was not feasible. As a reference, blood was collected from nine healthy adult volunteers.

For each sample, cfDNA fragment concentrations (>80 bp and >105 bp) were quantified using the qPCR assay described in Section 2.6, and PS values (0–100 scale) were calculated as described in Section 2.7. To assess the impact of non-cancer conditions, we computed the “PS Change” for each patient as the difference between their PS and the mean PS of the healthy cohort (baseline). A positive PS Change indicates a higher PS relative to healthy controls, potentially mimicking progression-like signals, while a negative or zero PS Change suggests minimal impact. Results are presented in Appendix A, with a footnote defining PS Change and its interpretation. This analysis provides an estimate of which clinical events may interfere with the PS assay’s specificity.

While comparing cfDNA from healthy volunteers to patients with acute conditions is a suboptimal approach to fully understanding the assay’s specificity, it highlights conditions likely to have the most significant impact, particularly those occurring between the first and second blood draws in cancer patients. Our findings suggest that conditions such as acute stroke, myocardial infarction, COPD exacerbation, and rheumatoid arthritis in patients treated with methotrexate may elevate cfDNA levels, potentially leading to false-positive PS results. However, this was a minimal study with only 2–5 patients per category, requiring further investigation. Ideally, a validation study would assess these conditions in treated cancer patients before blood draws, though practical limitations exist due to the rarity of these events in this population.

### 4.2. Limitations of This Study

This study demonstrates that cfDNA, combined with fragmentomics, can accurately measure changes in tumor burden in stage IV cancer patients. However, the study is preliminary and has several limitations. One limitation is the temporal gap between PS testing (12–21 days post-treatment) and radiographic imaging (8–12 weeks), which may lead to false negatives if tumor growth occurs after the second blood draw. Longitudinal blood collection, as in a prior study with MD Anderson [33], could address this by capturing dynamic cfDNA changes over time.

Another limitation is the potential for overfitting in the feature selection process, as the six pre-defined models were evaluated using the entire cohort (*n* = 128). To mitigate this, we used a hypothesis-driven approach based on prior cfDNA fragmentomic studies [32,33], selecting models grounded in biological rationale rather than purely data-driven methods, which may introduce spurious associations in small cohorts. Leave-one-out cross-validation (LOOCV) was employed to assess robustness, yielding a modest AUC reduction (0.934 to 0.880), suggesting limited overfitting. However, to confirm generalizability, we plan to validate the assay and algorithm in an independent cohort of stage IV cancer patients, which will include diverse cancer types (e.g., breast, colorectal, lung) and treatment regimens to ensure robust performance across clinical settings.

Additionally, the study was not designed to address imaging limitations related to pseudoprogression, which may contribute to false negatives, particularly in the four immunotherapy-treated patients with low PS values despite radiographic progression. A follow-up study with serial cfDNA measurements and clinical outcome correlations is needed to distinguish true progression from pseudoprogression.

Finally, this study did not evaluate the impact of the PS assay on clinical decision-making, patient quality of life, or treatment outcomes, which will be investigated in future prospective trials.

### 4.3. Opportunities for Improved Clinical Management

The PS assay presents four clear opportunities to enhance clinical management. First, it enables the rapid identification of patients experiencing disease progression, allowing clinicians to transition them to alternative therapeutic strategies promptly. Its real-time monitoring capability further permits the evaluation of low-probability treatment regimens after a single cycle, thereby providing early insight into patient responsiveness.

Second, there is a growing awareness of the overtreatment of stage IV cancer patients. In addition to the potential toxicological harm of a treatment for a patient’s health, there is the time and financial toxicity of cancer treatments. This PS assay may provide physicians with additional flexibility to adjust the treatment plan by removing agents or reducing dosing for a patient, or to give patients a break from treatment altogether. This can potentially reduce the level of toxicity and morbidity experienced by patients, thereby improving their quality of life and extending their overall survival. Two-arm randomized clinical studies could demonstrate that the PS assay enhances patient outcomes and quality of life while reducing the cost of cancer treatment. The ability to “de-escalate” treatment–to stop ineffective therapy without changing the survival of patients can improve quality of life and minimize financial toxicity. This is particularly important in LMICs (Low- and Middle-Income Countries), where patients often pay for treatment out of pocket.

Third, the PS assay alleviates both patient burden and clinical trial costs by swiftly identifying non-responders. Early recognition of ineffective treatment spares participants unnecessary exposure to therapy and reduces overall study expenses. Moreover, incorporating the PS assay into dose-finding and pharmacodynamic investigations provides real-time feedback, streamlining study design and accelerating decision-making in clinical trials.

Finally, immune checkpoint inhibitors (ICIs) have revolutionized cancer therapy by harnessing the patient’s own immune system to eradicate tumor cells. To date, the U.S. Food and Drug Administration has approved three classes of ICIs, targeting CTLA-4, PD-1/PD-L1, and LAG-3, with additional agents in development across over 20 tumor types in the neoadjuvant, adjuvant, and metastatic settings. Unlike cytotoxic therapies, ICIs can precipitate immune-related adverse events (irAEs) of varying severity (grades 1–4), which may occur at any point during or even after treatment, reflecting excessive immune activation. Moreover, unconventional response patterns such as pseudoprogression—an initial increase in tumor size followed by regression—and hyperprogression—accelerated disease growth seen in up to 30% of patients—pose significant clinical challenges, often leading to premature discontinuation of effective therapy and poorer outcomes [40,54]. By providing an early, dynamic readout of tumor-derived cfDNA changes, the PS assay could help distinguish actual progression from transient immune phenomena and identify non-responders before severe irAEs arise, thereby optimizing patient selection and improving the safety and efficacy of ICI treatment.

## 5. Conclusions

The PS assay could be a powerful tool, providing clinicians with valuable real-time data on disease progression and non-progression. This information can help optimize treatment plans and advance drug development studies. By combining cfDNA with fragmentomics, the PS assay can detect stage IV breast, colorectal, and lung cancer patients with disease progression as early as 12 days after beginning treatment. These findings demonstrate that, when paired with fragmentomic data, cfDNA could serve as an important cancer marker for evaluating tumor burden.

## Figures and Tables

**Figure 1 cancers-17-03583-f001:**
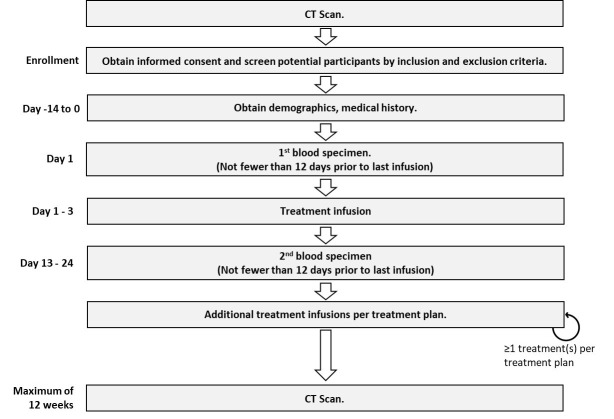
CADEX–0001 study flow diagram.

**Figure 2 cancers-17-03583-f002:**
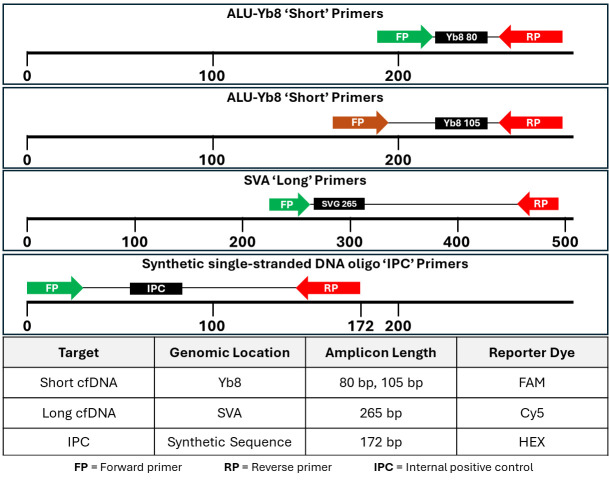
Schematic of primer and probe design for qPCR multiplexes.

**Table 1 cancers-17-03583-t001:** Six Pre-defined Models.

Model	Hypothesis
FragDiff	Change in tumor burden indicates changes in tumor size.
Frag1 + FragDiff	Initial tumor burden as measured by Frag1 adds predictive power.
SM1 + FragDiff	Initial tumor burden as measured by SM1 adds predictive power.
MMDiff + FragDiff	Add MMDiff to the above models. MMDiff is expected to refine FragDiff by helping to clean up noise (i.e., negative coefficient) from non-cancer sources of cfDNA.
Frag1 + MMDiff + FragDiff	
SM1 + MMDiff + FragDiff	

**Table 2 cancers-17-03583-t002:** Model Results.

Model	AUC	Note
FragDiff	0.802	
Frag1 + FragDiff	0.847	
SM1 + FragDiff	0.910	
MMDiff + FragDiff	*0.934*	Best model
Frag1 + MMDiff + FragDiff	0.936	Frag1 is not statistically significant.
SM1 + MMDiff + FragDiff	0.934	SM1 is not statistically significant.

**Table 3 cancers-17-03583-t003:** Progression Score Assay Result Interpretation.

Progression Score (PS)	n	Interpretation	Performance
≥90	11	Progression	PPV = 100%
<90	117	Likely non-progression	NPV = 92%

**Table 4 cancers-17-03583-t004:** A sample of participants with high levels of short and long cfDNA changes due to high levels of white blood cell lysis, and the impact of fragmentomics on assay performance. Considering only the change in concentration levels of >80 bp cfDNA fragments, versus the Progression Score.

Participant	PD by Imaging	Δ > 80 bp cfDNA	PS
2022	No	10.1-fold increase	8.6
4019	No	8.8-fold increase	0.0
8003	No	3.8-fold increase	0.1
2021	Yes	2.2% decrease	100.0

## Data Availability

All data and supporting results reported in this article, excluding confidential patient information, are available from the authors upon request.

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
