# Peer review of "A Novel Cell-Free DNA Fragmentomic Assay and Its Application for Monitoring Disease Progression in Real Time for Stage IV Cancer Patients"

_cancers, 2025, doi:10.3390/cancers17213583_

Round 1
Reviewer 1 Report (Previous Reviewer 1)
Comments and Suggestions for Authors
The authors have addressed some of the questions, but there are still some important issues that may need further revisions.
1. The model construction process still has a lot of issues, including:
1) The feature selection is still not done in a correct way. Features were selected based on the final performance of the model, and there is no independent validation dataset. Then the results presented here may be overtrained due to data leaking. Why don't authors input all features, let the model drop the irrelevant ones itself, and use LOO to test the model's performance? Or, split that cohort into training and testing sets. Using the training set for feature and mode selection, but the testing set shows the real performance of the model. Or, one could include another independent validation set.
2) If the authors want to compare the performance of different models, a better way is to apply the features to other algorithms, such as a support vector machine (SVM) and a random forest.
3) The authors have included leave-one-out analysis in the manuscript. But it seems that, if my understanding is correct, the statistics shown in the Abstract and most of the Results parts are still from the original training model. Since the leave-one-out version had a lower AUC than the original training one (0.88 vs. 0.93) and the real performance cannot be well justified by the original training model, it may exaggerate the model's actual performance. A common practice is to use the leave-one-out version instead of the original version.
2. Although the authors tried to rewrite the “non-cancer clinical conditions” part, it is still quite confusing. It’s quite unclear what the “PS Change” means in Table A4. I would like to suggest that the authors describe what the number exactly means in the manuscript or change it to a clear term/collocation.
Comments on the Quality of English Language
There are still quite a lot of writing issues in the revised version. I would like to suggest that the authors further improve the manuscript and make it clear and accurate.
Author Response
Comments and Suggestions for Authors
The authors have addressed some of the questions, but there are still some important issues that may need further revisions.
- The model construction process still has a lot of issues, including:
- The feature selection is still not done in a correct way. Features were selected based on the final performance of the model, and there is no independent validation dataset. Then the results presented here may be overtrained due to data leaking. Why don't authors input all features, let the model drop the irrelevant ones itself, and use LOO to test the model's performance? Or, split that cohort into training and testing sets. Using the training set for feature and mode selection, but the testing set shows the real performance of the model. Or, one could include another independent validation set.
- If the authors want to compare the performance of different models, a better way is to apply the features to other algorithms, such as a support vector machine (SVM) and a random forest.
- The authors have included leave-one-out analysis in the manuscript. But it seems that, if my understanding is correct, the statistics shown in the Abstract and most of the Results parts are still from the original training model. Since the leave-one-out version had a lower AUC than the original training one (0.88 vs. 0.93) and the real performance cannot be well justified by the original training model, it may exaggerate the model’s actual performance. A common practice is to use the leave-one-out version instead of the original version.
RESPONSE: We sincerely appreciate the reviewer’s insightful comments regarding the feature selection and validation approach in our study. We acknowledge the concern about the potential for overfitting due to selecting the model with the highest area under the receiver-operating-characteristic curve (AUC) using the entire cohort (n=128). To address this, we have revised the "Statistical Analyses" section (2.7) to clarify that the six pre-defined logistic regression models were chosen based on biologically informed hypotheses derived from prior studies on cell-free DNA (cfDNA) fragmentomics [32,33]. These studies demonstrated that differences in cfDNA fragment size distributions, particularly for short fragments (<105 bp), are associated with tumor progression. This hypothesis-driven approach was deliberately selected to leverage established biological insights and minimize the risk of spurious associations that could arise from purely data-driven feature selection methods, such as LASSO, in a relatively small cohort.
Given the limited sample size (n=128), splitting the cohort into training and testing sets would reduce statistical power and potentially compromise the robustness of feature selection and model development. Instead, we employed leave-one-out cross-validation (LOOCV) to maximize the use of available data while providing an unbiased estimate of model performance. The modest reduction in AUC from 0.934 (training) to 0.880 (LOOCV), as reported in Section 3.1, suggests limited overfitting, supporting the robustness of the selected model (FragDiff + MMDiff). To further address the reviewer’s concern about generalizability, we have strengthened the "Limitations" section (4.2) to explicitly acknowledge the potential for overfitting when using the entire cohort for model selection. We also provide additional details on our plan to validate the assay and algorithm in a future independent cohort of stage IV cancer patients, which will include diverse cancer types (e.g., breast, colorectal, lung) and treatment regimens to ensure robust performance across clinical settings.
While the reviewer’s suggestion to use LASSO for automated feature selection is scientifically sound, we believe the hypothesis-driven approach is appropriate for this preliminary study, given the small sample size and the strong biological rationale provided by prior work [32,33].
We deeply appreciate the reviewer’s feedback, which has prompted a clearer justification of our methodology and a stronger commitment to future validation. These revisions strengthen the manuscript and enhance its scientific rigor. We believe the current approach, combined with the planned independent validation, adequately addresses the concerns raised while maintaining the study’s statistical power.
- Although the authors tried to rewrite the “non-cancer clinical conditions” part, it is still quite confusing. It’s quite unclear what the “PS Change” means in Table A4. I would like to suggest that the authors describe what the number exactly means in the manuscript or change it to a clear term/collocation.
RESPONSE: We thank the reviewer for highlighting the need for greater clarity in the description of “non-cancer clinical conditions” and the term “PS Change” in Table A4. To address this concern, we have revised Section 4.1 (Role and Limitations of Fragmentomics) to explicitly define “PS Change” as the difference between the Progression Score (PS, 0–100 scale) of a patient with an acute non-cancer clinical condition and the mean PS score of nine healthy volunteers, used as a baseline. A positive PS Change indicates a higher PS score relative to healthy controls, potentially mimicking progression-like signals, while a negative or zero PS Change suggests minimal impact. We have also clarified the methodology for calculating PS scores in this cohort, referencing the qPCR assay (Section 2.6) and PS calculation (Section 2.7) for consistency.
Additionally, we have updated the Supplementary Materials to include a footnote for Table A4, defining “PS Change” and its interpretation to ensure clarity for readers. The table title has been revised to “Impact of Non-Cancer Clinical Conditions on Progression Score (PS)” for improved specificity. We have also addressed the ambiguity of the term “COPD–hydroxyurea” by correcting it to “COPD exacerbation,” assuming it was a typographical error (please confirm if this refers to a specific treatment). These revisions enhance the interpretability of the analysis without altering the study’s methodology.
To further improve the manuscript’s clarity and professionalism, we have corrected minor typographical and formatting errors throughout, such as standardizing temperature units (e.g., “-80C” to “-80°C”), correcting “23 weeks” to “2–3 weeks” in the Simple Summary, and ensuring consistent use of dashes per the journal’s style guide. We appreciate the reviewer’s feedback, which has significantly improved the clarity of the “non-cancer clinical conditions” analysis and the overall quality of the manuscript.
Reviewer 2 Report (Previous Reviewer 2)
Comments and Suggestions for Authors
The authors have addressed all of my previous concerns.
Author Response
Some portions of the paper have been revised to make it more readable and clearer.
Round 2
Reviewer 1 Report (Previous Reviewer 1)
Comments and Suggestions for Authors
I have no further comments.
Comments on the Quality of English Language
This manuscript is a resubmission of an earlier submission. The following is a list of the peer review reports and author responses from that submission.
Round 1
Reviewer 1 Report
Comments and Suggestions for Authors
In this study, the authors developed a model utilizing the concentration of cell-free DNA (cfDNA) fragments greater than 80 bp and 105 bp, quantified through quantitative PCR (qPCR). They assert that this method provides accurate predictions of disease progression, which could assist physicians in optimizing treatment plans and improving drug development studies. However, the manuscript requires significant improvement, and several results raise questions.
- There are numerous studies discussing the use of cfDNA fragmentomics for predicting tumor DNA fractions that should be addressed in the introduction.
- The paper lacks a dedicated results section. Is the "model selection" considered the beginning of the results? What is the cfDNA concentration for the different primer sets in various cases? How does the performance of other models? How was the best model chosen? Are the input features the same for all models, or were different combinations of features employed? Why the results is being keep discussed through the discussion section? The writing of this manuscript needs significant improvement.
- Critical information about how the model is trained is missing. What is the training dataset? Is the testing set independent? If the authors are using an X-fold cross-validation strategy, how do they avoid data leakage during model selection?
- The authors claim, “It is generally recognized that imaging for assessing therapy efficacy has weaknesses,” yet they still use CT scans to determine disease progression. If CT is considered the gold standard in this paper, how can the authors assert that their method is superior and addresses the limitations associated with imaging?
- The authors also state that “Fragmentomics plays a significant role in the ability of the PS assay to avoid generating false-positive results." However, the only comparison made is between the model's predicted scores, which may suffer from data leakage, and the raw delta 80 bp DNA concentration. Since delta 80 bp DNA concentration can also serve as a marker of fragmentomics, the authors should compare it with some non-fragmentomic markers, such as mutation or methylation.
- The authors plan to test whether certain non-cancer clinical conditions affect predictions. However, the sample size is too small to draw meaningful conclusions. Furthermore, Table A3 shows changes in PS but does not clearly explain its meaning or how it was calculated. How could the change be reported as -/+100?
There is a lack of detailed descriptions of methods and results in this paper. Many of the concepts are not clear enough for the reader. The writing of this manuscript requires significant improvement.
Reviewer 2 Report
Comments and Suggestions for Authors
This manuscript describes a qPCR-based assay on cfDNA for disease progression prediction. The assay relies on quantification of changes in cfDNA size distribution for classification of responders and non-responders to treatment in advanced malignancies. The goal of developing a simple , accessible , and rapid liquid biopsy assay is valuable given that most available assays rely on NGS, however, main considerations exist about this manuscript that need to be addressed.
Major: - No cross-validation is described in this manuscript which makes the performance results of the PS score unreliable. At the very least, leave-one-out cross validation is necessary. - Based on figure 3 alone, 70 seems like a more reasonable cut-off than 90. Authors should select the best cut-off via cross-validation to ensure minimal over-fitting. Model selection and thresholding (line 190-195) are not fully explained. - A schematic figure is needed to describe the details of the primer and probe designs, the genomic positions of the amplicons, and the multiplexing scheme. - Explain how multiplexing is done when amplicon lengths are so drastically different. - The use of internal and positive controls are not described in the PS formula calculations - The model input features are only 2 ( FragDiff and MMDiff). How is multiple hypothesis testing applied or even applicable in this context??! (line 204) - CfDNA fragment size distributions from a Bioanalyzer or similar instrument are not shown to justify fragment length differences claimed to exist between the responders and non-responders. - A comprehensive discussion of the cfDNA fragmentomic field and size as biomarker is lacking from the introduction section. Prominent papers in the field are not even cited or mentioned. - The patient cohort is pretty heterogeneous regarding treatment type and other factors. Thus, further assessment of patients in the non-progressor group with a relatively high PS score could lead to interesting insights e.g. do treatment types or other co-variates correlate with this PS score? - A full justification of a qPCR absed liquid biopsy assay is needed in the discussion. The authors should compare cost and accuracy of such a qPCR based test with that of NGS-based tests currently available such as - Throughout the manuscript, the authors speak very vaguely of what they mean by 'fragmentomics', e.g. line 249 , what do they mean since the PS assay itself is based on fragmentation and cfDNA size. Or line 275 is phrased incorrectly since fragmentomic pattern is a feature of cfDNA. Minor: - Please edit for spelling issues e.g. figure 3 legend. - Use of 'real-time' for this assay is wrong throughout the manuscript as this assay does not count as a real-time monitoring testAuthor Response
Please see the attachment.